# Genome-Wide Identification and Characterization of the Soybean Snf2 Gene Family and Expression Response to Rhizobia

**DOI:** 10.3390/ijms24087250

**Published:** 2023-04-14

**Authors:** Jianhao Wang, Zhihui Sun, Huan Liu, Lin Yue, Fan Wang, Shuangrong Liu, Bohong Su, Baohui Liu, Fanjiang Kong, Chao Fang

**Affiliations:** Guangzhou Key Laboratory of Crop Gene Editing, Guangdong Key Laboratory of Plant Adaptation and Molecular Design, Innovative Center of Molecular Genetics and Evolution, School of Life Sciences, Guangzhou University, Guangzhou 510006, China; sun3020009@163.com (Z.S.); lh@gzhu.edu.cn (H.L.); yuelin@gzhu.edu.cn (L.Y.); wangfan@gzhu.edu.cn (F.W.); liusr@gzhu.edu.cn (S.L.); subohong@gzhu.edu.cn (B.S.); liubh@gzhu.edu.cn (B.L.); kongfj@gzhu.edu.cn (F.K.)

**Keywords:** *Glycine max*, ATP-dependent chromatin remodeling, Snf2 gene family, nodulation, rhizobia

## Abstract

Sucrose nonfermenting 2 (Snf2) family proteins are the core component of chromatin remodeling complexes that can alter chromatin structure and nucleosome position by utilizing the energy of ATP, playing a vital role in transcription regulation, DNA replication, and DNA damage repair. Snf2 family proteins have been characterized in various species including plants, and they have been found to regulate development and stress responses in Arabidopsis. Soybean (*Glycine max*) is an important food and economic crop worldwide, unlike other non-leguminous crops, soybeans can form a symbiotic relationship with rhizobia for biological nitrogen fixation. However, little is known about Snf2 family proteins in soybean. In this study, we identified 66 Snf2 family genes in soybean that could be classified into six groups like Arabidopsis, unevenly distributed on 20 soybean chromosomes. Phylogenetic analysis with Arabidopsis revealed that these 66 Snf2 family genes could be divided into 18 subfamilies. Collinear analysis showed that segmental duplication was the main mechanism for expansion of Snf2 genes rather than tandem repeats. Further evolutionary analysis indicated that the duplicated gene pairs had undergone purifying selection. All Snf2 proteins contained seven domains, and each Snf2 protein had at least one SNF2_N domain and one Helicase_C domain. Promoter analysis revealed that most Snf2 genes had *cis*-elements associated with jasmonic acid, abscisic acid, and nodule specificity in their promoter regions. Microarray data and real-time quantitative PCR (qPCR) analysis revealed that the expression profiles of most Snf2 family genes were detected in both root and nodule tissues, and some of them were found to be significantly downregulated after rhizobial infection. In this study, we conducted a comprehensive analysis of the soybean Snf2 family genes and demonstrated their responsiveness to Rhizobia infection. This provides insight into the potential roles of Snf2 family genes in soybean symbiotic nodulation.

## 1. Introduction

Chromatin, which consists of nucleosomes as its basic units, is the eukaryotic DNA-protein complex that allows efficient packaging of large genomes within nuclei. Nucleosomes are composed of 147 base pairs of DNA that are tightly wrapped around an octamer of histone proteins, comprised of two copies each of H2A, H2B, H3, and H4 [1]. However, chromatin also poses a barrier for regulatory factors such as transcription factors to access DNA. Therefore, eukaryotes have evolved precise mechanisms to regulate chromatin structure and dynamics in a spatiotemporal manner, which is essential for gene expression control. These mechanisms include covalent modification of DNA and histones, replacement of histone variants, regulation of nucleosome assembly by histone chaperones and ATP-dependent chromatin remodeling mediated [2,3,4]. The Sucrose nonfermenting 2 (Snf2) family proteins are the core component of chromatin-remodeling complexes that alter DNA accessibility by various modes such as dimer exchange, nucleosome spacing, nucleosome eviction or nucleosome sliding [5].

Chromatin-remodeling complexes regulate gene transcription by altering chromatin structure and accessibility. They are classified into four groups: SWI/SNF(switch/sucrose nonfermentable), CHD (chromodomain, helicase, and DNA binding), INO80 (inositol requiring 80) and ISWI (imitation switch) [6]. Each group has a distinct protein composition and function. The SWI/SNF complex regulates gene transcription by changing the position of nucleosomes at the promoter region [7]. The ISWI complex regulates gene transcription by assembling and arranging nucleosomes [8]. The INO80 complex is able to bind to phosphorylated H2AX or γ-H2AX, allowing for the recruitment of repair factors to double-stranded breaks for DNA repair [9,10]. CHD complexes can recognize H3K4me3 and regulate transcriptional elongation and splicing through interaction with other preinitiation factors. They can also regulate transcription by influencing histone modifications through complex formation with other epigenetic factors [11,12,13].

The Snf2 protein name derives from the *Saccharomyces cerevisiae* Snf2 protein [14]. The Snf2 family proteins contains two conserved domains: the SNF2-N and helicase-C domains [15,16]. They are classified into six groups based on similarity within the helicase-like region and functional differences: Snf2-like, Swr1-like, Rad54-like, SSO1653-like, Rad5/16-like, and SMARCAL1-like [15,16]. These six groups include 24 distinct subfamilies that are widely conserved across eukaryotes [3]. Snf2 family proteins are conserved and diversified ATPases that participate in chromatin remodeling in various plant species. The Snf2 gene family is well-represented in the genomes of several plant species, with 41 members in *Arabidopsis thaliana*, 40 in *Oryza sativa*, 45 in *Solanum lycopersicum*, and 38 in *Hordeum vulgare*. [15,17,18,19]. Snf2 genes regulate various plant processes related to growth and abiotic stress responses [20,21,22,23]. Flowering is an important process in plant life that directly affects yield. Several Snf2 proteins in Arabidopsis regulate flowering [24,25]. Another process is plant organ development, which involves Snf2 family genes. For example, BRAHMA (BRM) regulates root length and floral organ identity [26,27], SPLAYED (SYD) controls shoot apical meristem (SAM) development [28], and CHROMATIN REMODELING 11/17 (CHR11/17) influences flower organ development [29]. Moreover, Snf2 family genes are often involved in hormone response processes, including auxin, gibberellin (GA), cytokinin (CK), abscisic acid (ABA), and brassinosteroid (BR) signaling pathways. For example, BRM affects auxin distribution by influencing the the expression levels of *PIN-FORMED (PIN)* genes; the Cytokinin-Hypersensitive2 (CKH2)/PICKLE (PKL) chromatin remodeling factor negatively regulates cytokinin responses in Arabidopsis calli [26,30,31,32,33]. Similarly, Snf2 family genes play crucial roles in plant responses to various environmental stresses, including heat, drought, salt, and cold [20]. For instance, BRM plays a crucial role in plant responses to heat stress, drought, and salt stress [30,34,35,36], while PKL responds to cold and salt stress [37]. CHR5 responds to pathogen attack [38].

Soybean (*Glycine max*) is a major crop worldwide due to its high protein and edible oil content [39,40,41]. Unlike other non-leguminous crops, soybeans can form a symbiotic relationship with rhizobia for biological nitrogen fixation, which converts atmospheric nitrogen into ammonia as a nitrogen source for plant growth [42]. The soybean root secretes flavonoid compounds to induce the synthesis and secretion of Nod factors (NFs) by rhizobia. These NFs are recognized by NF receptors such as Nod factor Receptor (NFR1ɑ/NFR5ɑ) in soybean, which activate the expression of nodulation genes, including transcription factors *Nodulation Signaling Pathway (NSP1/NSP2)*, *Nodule Inception (NIN)*, and *Early Nodulin40 *(*ENOD40*). This leads to the formation of rhizobial primordia [43,44,45,46]. Rhizobia-Induced CLE1 (RIC1) and RIC2, move from the roots to aboveground tissues through the xylem and are recognized by the receptor kinase NODULE AUTOREGULATION RECEPTOR KINASE (NARK) in the leaves blade through an autoregulation of nodulation pathway [47,48]. Moreover, factors produced by aboveground tissues, including cytokinins, miRNA2111, and others, move through the phloem to the roots, where they interact with Too Much Love (TML), NIN, and other factors to regulate the expression of *CLAVATA3/endosperm-surrounding region* (*CLE*) and control root nodulation [49,50,51]. Soybean nodulation and nitrogen fixation involve a complex regulatory network. Although some regulatory genes have been discovered in the past 20 years, many aspects of this network remain unclear and many genes await discovery. Chromatin remodeling is an important epigenetic regulation component that has been rarely reported in soybean nodulation research. Recent studies have found that histone deacetylases (HDACs) in Medicago truncatula affect primordium formation by regulating *3-hydroxy-3-methylglutaryl coenzyme a reductase 1 (MtHMGR1) gene expression* [52]. The Snf2 family genes, an important gene family in chromatin remodeling, have been rarely studied in soybean. To better understand the role of chromatin remodeling in nodulation, this study provides a comprehensive analysis of the soybean Snf2 family genes and their gene expression. It is suggested that Snf2 family genes may participate in soybean symbiotic nodulation. This provides insight into the potential roles of Snf2 family genes in soybean symbiotic nodulation.

## 2. Results

### 2.1. Identification of Soybean Snf2 Family Proteins

In order to comprehensively identify all Snf2 family proteins in soybean, we conducted a search using a Hidden Markov Model (HMM) and the SNF2_N (Pfam: PF00176) and Helicase_C (Pfam: PF00271) domains as queries against the soybean protein database available on Phytozome This resulted in 502 genes encoding proteins with the conserved Helicase_C domain and 167 genes encoding proteins with the SNF2_N domain. The accuracy of these proteins was further verified using CDD and SMART. Finally, we identified 66 high-confidence genes encoding proteins containing both the Helicase_C and SNF2_N domains as members of the soybean Snf2 family (Appendix A). Following the nomenclature used for rice and Arabidopsis, we named the soybean Snf2 family proteins GmCHRs. 

In order to examine the evolutionary relationship between Snf2 family proteins in soybean and Arabidopsis, the MEGAX software was used to analyze the Snf2 family members in both species using the maximum likelihood method. The results showed that the soybean Snf2 family proteins were classified into six groups and eighteen subfamilies. (Figure 1). Most of the soybean Snf2 family members were closely related to their Arabidopsis counterparts and were distributed across all the subfamilies, with more members in the Snf2, DRD1, ERRC6, and Rad5/16 subfamilies (Figure 1). Among these subfamilies, the Snf2 subfamily had the most members, with 9 members; while the ALC1 and Rad54 subfamilies had only one member each (Figure 1 and Appendix A).

### 2.2. Analysis of Chromosomal Distribution and Duplication of the Soybean Snf2 Family Genes

Using TBtools to draw the chromosome localization map of the soybean Snf2 family genes. We mapped *GmCHRs* to the soybean genome and discovered an uneven distribution of 66 Snf2 family genes across the 20 chromosomes. The majority of these genes were located near the chromosomal ends, while a few were situated in the middle regions of the chromosomes (Figure 2). The number of Snf2 genes on each chromosome also varies. There is only 1 Snf2 gene on chr06, chr14, and chr19; 2 on chr03, chr04, chr05, chr15, and chr18; 3 on chr01 and chr11; 4 on chr02, chr08, chr09, and chr17; 5 on chr07 and chr20; 6 on chr10 and chr12; and 8 on chr13 (Figure 2).

Tandem and segmentalduplication are widely acknowledged as crucial mechanisms for the expansion of gene families [53]. Therefore, we conducted a survey to determine segmental duplication in the formation of the soybean Snf2 family genes. In general, tandem duplication is characterized by the presence of two paralogous genes located in close proximity to each other on the same chromosome, with typically no more than 5 intervening genes separating them. We detected no instances of tandem duplication, but the analysis revealed 34 pairs of 41 genes each that likely underwent segmental duplication (Figure 3). These results indicate that the origin of the Snf2 family genes in soybean is likely attributable to segmental duplication rather than tandem duplication. Similarly, Snf2 family genes in rice and Arabidopsis were found to have experienced only segmental duplication events [18]. Next, we computed the ratio of nonsynonymous (Ka) to synonymous (Ks) substitutions (Ka/Ks) to explore the possible selective pressure driving the duplication events of the *GmCHRs* (Appendix A). Ka/Ks ratios are considered to indicate purifying selection, indicating that natural selection has removed deleterious mutations and maintained protein stability; Ka/Ks ratios less than 1 are considered to indicate purifying selection, indicating that natural selection has removed deleterious mutations and maintained protein stability; Ka/Ks ratios greater than 1 indicate positive selection, suggesting that natural selection has acted on changes in the protein, causing the mutated sites to rapidly fix in the population and accelerate the evolution of the gene; Ka/Ks ratios equal to 1 indicate neutral selection, suggesting that natural selection has no effect on the mutation [54]. The 34 pairs of genes that underwent segmental duplication all exhibited Ka/Ks ratios lower than 1 (0.056–0.506), indicating that the duplicated genes were subjected to purifying selection pressure (Appendix A).

### 2.3. Analysis of Gene Structure and Conserved Domains in the Soybean Snf2 Family

We analyzed the Snf2 family proteins and found that they have amino acid numbers ranging from 276 to 3789, molecular weights (Mw) ranging from 32.1 to 410.7 kDa, and theoretical isoelectric points (pI) ranging from 4.96 to 9.3 (Appendix A). In order to gain a better understanding of the functions of Snf2 family proteins, the conserved domains of Snf2 proteins were analyzed using the Conserved Domain Database (CDD) and Pfam websites. The analysis revealed that the Chromo domain was only found in the Chd1 and Mi-2 subfamilies of the soybean Snf2 genes (Figure 4A). The PHD domain was present in all members of the Mi-2 subfamily except for GmCHR58 (Figure 4A). Interestingly, GmCHR5 from the SHPRH subfamily also contained the PHD domain (Figure 4A). The zf-C3HC4 domain was detected in almost all members of the SHPRH, Rad5/16 and Ris1 subfamilies, except for GmCHR36, GmCHR11 and CmCHR28 (Figure 4A). However, the F-box domain was only observed in GmCHR36 and GmCHR11 from the SHPRH subfamily (Figure 4A).

Analysis of exon/intron structures of 66 Snf2 family genes showed that the number of exons varied greatly among Snf2 family members, ranging from 36 (*GmCHR55*) to 2 *(GmCHR29*) (Figure 4B). Further analysis revealed that the members of DRD1 subfamily had significantly fewer exons than those of other subfamilies (Figure 4B). The Snf2 family genes usually had very long sequences, with 13 genes exceeding 20 kb and 3 genes exceeding 30 kb (Figure 4B).

### 2.4. Analysis of Cis-Element the Soybean Snf2 Gene Promoters

Because gene transcription regulation is typically achieved through the binding of different transcription factors to *cis*-elements in the promoter. To explore the transcription regulation of Snf2 genes in response to various environmental signals, we analyzed the 2 kb promoter regions of 66 soybean Snf2 family genes using the PlantCARE. (Figure 5 and Appendix A) A total of 23 *cis*-elements were discovered in the promoter regions of soybean Snf2 family genes. Five of these *cis*-elements are related to hormones, such as methyl jasmonate (MeJA), abscisic acid (ABA), gibberellin (GA), auxin, and salicylic acid (SA). These hormone-related *cis*-elements are widely distributed in the promoter regions of Snf2 family genes, especially ABA- and MeJA-responsiveness *cis*-elements (Figure 5 and Appendix A). Some *cis*-elements related to stress response, such as drought, low temperature, wound, and tissue-specific expression (seed and root), are also widely distributed in the promoter regions of various genes. The distribution of these hormone- and stress-related *cis*-elements among different subfamilies does not seem to follow any specific pattern (Figure 5 and Appendix A). In addition, 12 types of *cis*-elements were related to metabolism regulation and development. (Figure 5 and Appendix A). It is worth noting that nodule specificity *cis*-element (5’AAAGAT) [55] is the second most widely distributed *cis*-element after MeJA responsiveness (17%) and ABA responsiveness (16%). Nodule specificity *cis*-element is distributed in the promoter regions of 52 Snf2 family genes, which suggests that these genes may be related to symbiotic nodulation. (Figure 5 and Appendix A).

### 2.5. Expression Profiles of the Snf2 Family Genes in Symbiotic Nitrogen Fixation

To explore the potential functions of the Snf2 genes in soybean, we obtained their expression patterns from the eFP Browser for soybean, an online transcriptome database. The expression patterns of Snf2 family genes were analyzed in root and nodule (Figure 6A and Appendix A). However, the database did not contain information on the remaining 14 genes. To examine how Snf2 family genes respond to rhizobial infection, the expression profiles of 52 Snf2 genes in root hairs at 12 and 24 h after inoculation (HAI) were analyzed. Figure 6A shows that out of the 52 Snf2 genes, more than 50% had higher expression levels in nodule than in root. Furthermore, the expression of Snf2 genes in infected root hairs at different time points after inoculation was analyzed. The heatmap shows that these Snf2 genes were responsive at 12 HAI in infected root hairs, but they responded differently (Figure 6B and Appendix A). Some genes (such as *GmCHR16*, *GmCHR35* and *GmCHR51*) were upregulated after inoculation, while some genes (such as *GmCHR30*, *GmCHR9* and *GmCHR15*) were downregulated after inoculation (Figure 6B and Appendix A). Overall, there were more upregulated than downregulated genes. Interestingly, most of the genes with large expression differences at 12 HAI showed smaller differences at 24 HAI. For example, *GmCHR4, GmCHR24*, *GmCHR44* and other genes were downregulated at 24 HAI; whereas *GmCHR26*, *GmCHR18* and *GmCHR27* and other genes were upregulated at 24 HAI. These genes had different expression patterns at 12 HAI and 24 HAI (Figure 6B and Appendix A). Although some genes increased and some decreased in expression, almost all Snf2 genes responded to rhizobial infection, suggesting that Snf2 family genes may play important roles in symbiotic nitrogen fixation. 

To further analyze the potential role of Snf2 family genes in symbiotic nodulation, we performed qPCR analysis on 26 Snf2 genes containing nodule specificity *cis*-element and with Reads Per Kilobase per Million mapped reads (RPKM) greater than 2.5 in nodules in the eFP Browser. These genes were detected in roots or nodules at 28 days after inoculation (DAI). The results showed that all genes except *GmCHR5* were significantly more highly expressed in mature nodules than in roots (Figure 7A). To verify the response of these genes to rhizobial infection, we analyzed their expression induced by rhizobia at 24 h after inoculation (HAI) in root hairs using *GmNIN1a* and *GmENOD40.1* as positive controls. The results showed that both marker genes were upregulated after rhizobial infection. Nine genes did not show significant changes in expression, and the remaining genes were significantly downregulated (Figure 7B). The discrepancy between our results and microarray data may be caused by differences in plant culture or rhizobial infection efficiency. 

## 3. Discussion

The Snf2 family proteins are essential for chromatin-remodeling complexes that regulate transcription, replication, homologous recombination, and DNA repair in all eukaryotes. These proteins have diverse functions in plant development and stress responses. [6,20,56]. For example, Snf2 proteins are involved in flowering, organ formation, and stress response in Arabidopsis [20,21]. Previous studies on Snf2 family proteins have mainly focused on model plants such as Arabidopsis and rice. However, soybean is an important crop that differ from these non-leguminous crops due to its unique plant-rhizobia symbiosis. The role of Snf2 proteins in soybeans, especially in nodulation, remains unknown. Here, we identified 66 soybean Snf2 proteins (Appendix A). In this study, we identified 66 Snf2 proteins in soybeans and performed a comprehensive analysis of their phylogenetic relationships, gene classification, chromosomal locations, conserved domains, gene structures, *cis*-elements, and expression profiles in different tissues and during nodulation. Our results provide valuable insights into the Snf2 family proteins and highlight their potential functions in the symbiotic interaction between soybeans and rhizobia.

Soybeans contain 66 Snf2 family genes, which is more than Arabidopsis (41), rice (40), and tomato (45) [15,17,19], may be due to the partial diploidization of the tetraploid soybean genome, resulting in a higher number of Snf2 genes than in diploid species [57]. Phylogenetic analyses using Snf2 proteins from Arabidopsis and soybean classified the 66 soybean Snf2 proteins into 6 groups and 18 subfamilies (Figure 1). The number of Snf2 proteins in each subfamily ranged from 1 to 9 (Figure 1). Interestingly, the ALC1, Rad54, and SMARCAL1 subfamilies showed a 1:1 orthologous pattern between Arabidopsis and soybean, indicating that these subfamilies are more conserved than others (Figure 1). This is noteworthy given that genome duplications occurred around 59 and 13 million years ago, leading to a highly duplicated genome with nearly 75% of genes present in multiple copies [58]. We identified 34 pairs of segmental duplications in soybean, but no tandem duplications (Figure 3), which is similar to Arabidopsis and rice [18]. Notably, some segmentally duplicated pairs in Arabidopsis are functionally redundant (*CHR12/23* and *CHR11/17*) [59,60], and their homologs in soybean also had segmental duplication. This implies that the gene pairs *GmCHR18/56*, *GmCHR18/53*, *GmCHR32/65*, and *GmCHR34/65* may have similar functions. Therefore, segmental duplication, rather than tandem duplication, seems to be the main evolutionary mechanism for the expansion and functional diversification of the Snf2 gene family. The evolution of new gene functions usually results from the combined effects of duplication and selection. Our analysis found that all 34 segmental-duplication gene pairs had Ka/Ks ratios less than 1 (Appendix A), indicating that they underwent purifying selection and reduced genetic diversity. This implies that the functional divergence of these duplicated genes might tend to be conservative. 

The catalytic ATPase domain of Snf2 proteins is responsible for chromatin-remodeling activity. It consists of SNF2_N, which has ATP hydrolysis activity, and Helicase_C, which has ATP-dependent DNA or RNA unwinding activity. This structure is conserved in plants [15,16]. In soybean, besides these two conserved domains, each family also contains some unique domains. Proteins in the Mi-2 subfamily (except for GmCHR58) and SHPRH subfamily (only GmCHR5) have a PHD domain at their N-terminus (Figure 4). The PHD domain is a Zn^2+^-binding domain that can recognize and bind H3K4me3 in various proteins such as BPTF, YNG1 and ING2 [61,62,63,64]. In addition to H3K4me3, this domain can also recognize various other histone modifications, such as H3K9me3 recognition by the PHD domain of human Mi-2 homolog CHD4 [65]. This suggests that soybean proteins with PHD domains may crosstalk with other histone modifications. In both Chd1 and Mi-2 subfamily proteins, there is a Chromo domain upstream of Helicase_C (Figure 4). The CHROMO domain was first discovered in animals and has DNA-binding activity [66]. The CHROMO domain of rice Mi-2 subfamily protein OsCHR729 was also found to bind methylated H3K4, suggesting that these proteins with CHROMO domains may also have the ability to bind methylated H3K4 [67]. GmCHR36 and GmCHR11 from the SHPRH subfamily contain an F-box domain (Figure 4). Proteins containing an F-box domain are usually subunits of SCF complexes, which are E3 ubiquitin ligases that mediate the proteasomal degradation of specific substrates. F-box proteins function as substrate recognition components in SCF complexes [68]. This suggests that GmCHR36 and GmCHR11 may have E3 ubiquitin ligase activity. Moreover, most members of the three subfamilies in the Rad5/16-like group (SHPRH subfamily, Rad5/16 subfamily, Ris1 subfamily) have a zf-C3HC4 domain at their N-terminus (except for GmCHR11 and GmCHR36) (Figure 4). The zf-C3HC4 domain is a zinc-finger domain that may be involved in both DNA-binding and protein–protein interaction functions [69]. This implies that soybean proteins with zf-C3HC4 domains may play important roles in connecting other soybean chromatin-remodeling complex subunits.

The promoters of almost all Snf2 genes were found to contain diverse *cis*-elements, which are involved in organ development, plant hormone (such as abscisic acid and jasmonic acid) response and stress (drought, low temperature) tolerance (Figure 5). Notably, the *cis*-elements associated with jasmonic acid, abscisic acid, and nodule specificity were the most common (Figure 5 and Appendix A). These two hormones are known to be involved in the response to external biotic and abiotic stresses. The frequent occurrence of these three *cis*-elements in Snf2 gene promoters suggests a close relationship between Snf2 family genes and plant stress response. It also implies an essential role of Snf2 family genes in symbiotic nodulation. Soybean is a globally important food and economic crop, and the identification of these stress-related genes is crucial for developing new stress-resistant varieties.

Nitrogen is a crucial factor that limits crop growth and productivity. Soybean can form nodules by interacting with rhizobia in the soil to meet its own nitrogen demand [25]. The efficiency of soybean nodulation is determined by three main factors: nodule infection rate, nodule development process and nitrogen fixation capacity of mature nodules [26]. Chromatin remodeling is one of the major mechanisms of transcriptional regulation, which involves Snf2 family proteins as the core components of ATP-dependent chromatin remodeling complexes. These proteins may be involved in various aspects of plant life, including nodulation. For instance, Arabidopsis BRM regulates root development by affecting *PIN-FORMED* (*PIN*) expression and auxin distribution. Recent studies have demonstrated that *GmPINs* also regulate nodulation [27]. This implies that Snf2 family genes may have a significant role in nodulation as well. We examined the expression patterns of Snf2 genes in roots and nodules (Figure 7A). Many genes exhibited remarkable expression changes before and after rhizobial inoculation (such as *GmCHR34*, *GmCHR10*, *GmCHR66*, *GmCHR14*), while some genes had striking expression differences between roots and nodules (such as *GmCHR18*, *GmCHR19*, *GmCHR56* and *GmCHR66*) (Figure 7B). These findings suggest that these genes can respond to rhizobial infection and may participate in nodulation regulation by Snf2 family proteins. These genes could offer valuable resources for molecular breeding of high-efficiency nitrogen fixation.

## 4. Materials and Methods

### 4.1. Plant Materials and Growth Conditions

In this study, soybean Williams 82 (W82) was utilized as the plant material.The soybean seeds were germinated in sterile water for a period of five days before being transferred to vermiculite that was supplemented with a low-nitrogen culture solution. The soybeans were then grown in a greenhouse under controlled conditions, including a temperature of 25 °C, 70% humidity, and a photoperiod consisting of 16 h of light and 8 h of darkness for three days The rhizobial strain *Bradyrhizobium diazoefficens* USDA110 was inoculated at an optical density (OD) of 0.08 using sterile water as a carrier. 

### 4.2. Identification of the Soybean Snf2 Family Genes

To thoroughly identify all proteins within the soybean Snf2 family, a search was conducted for a high-confidence soybean genome in Phytozome 13. (https://phytozome.jgi.doe.gov/pz/portal.html, accessed on 23 November 2022), because all Snf2 proteins contain two conserved domains: SNF2_N and Helicase_C. The Hidden Markov Model (HMM) files for SNF2_N and Helicase_C were downloaded from the Pfam database (SNF2_N Pfam: PF00176, Helicase_C Pfam: PF00271). The Simple HMM Search plugin from TBtools (a software suite for biological data analysis) was used to retrieve 66 high-confidence genes that contained both SNF2_N and Helicase_C domains in W82 reference genome [70]. The candidates for Snf2 proteins were then confirmed using SMART (http://smart.embl-heidelberg.de/, accessed on 23 November 2022) and CDD (https://www.ncbi.nlm.nih.gov/Structure/cdd/wrpsb.cgi, accessed on 23 November 2022). The conserved motif in Snf2 proteins was detected through CDD, and a visual representation of the motif map was generated using TBtools [70]. 

### 4.3. Phylogenetic Construction

A phylogenetic tree file was generated by comparing the amino acid sequences of soybean and Arabidopsis Snf2 family proteins with the Maximum Likelihood method in MEGAX software (Molecular Evolutionary Genetics Analysis) [71]. The generated phylogenetic tree was then refined and beautified using the Evoview (http://www.evolgenius.info/evolview/, accessed on 23 November 2022).

### 4.4. Chromosome Localization, Duplication, and Evolution 

The chromosomal location of each Snf2 gene was determined by using a GTF file of soybean genome and an ID list of CHR family members with TBtools [70]. Synteny analysis on internal CHR genes in soybean was conducted with One Step MCScanX plugin in TBtools, and visualization was performed with Advanced Circos plugin [70]. Nonsynonymous (Ka) and synonymous (Ks) substitution rates for each Snf2 gene pair were calculated with Simple Ka/Ks Calculator in TBtools [70].

### 4.5. Characterization of Snf2 Family Proteins and Gene Structure

Theoretical grand average of hydropathicity (GRAVY), isoelectric point (pI) and molecular weight (MW) of soybean Snf2 proteins were analyzed using ProParam software on the ExPASy server (Expert Protein Analysis System), NCBI’s CDD platform (Conserved Domain Database) was used to predict conserved domains for all identified soybean Snf2 proteins’ polypeptide sequences. Data were then visualized using TBtools [70]. The exon-intron architecture diagrams were also created using TBtools with its Gene Structure View (Advanced) [70].

### 4.6. Promoter Analysis 

The 2 kb upstream of the translation initiation site of each Snf2 gene was defined as its promoter region, and PlantCARE was used to analyze these regions. The results were visualized with TBtools [70].

### 4.7. Expression Profile Analysis

Gene expression data of soybean Snf2 genes were retrieved from the Glycine max eFP Browser website and imported into TBtools to generate a heat map displaying the expression levels of soybean Snf2 genes in root and nodule, as well as in root hairs at 12/24 h after inoculation (HAI) [70].

### 4.8. RNA Isolation and Real-Time Quantitative RT-PCR Expression Analysis 

Total RNA from root materials was extracted using FastPure Cell/Tissue Total RNA Isolation Kit V (No.RC112-01, Vazyme). mRNA was reverse transcribed using TransScript Uni All-in-One First-Strand cDNA Synthesis SuperMix (No. AU341-02, Transgen). Gene expression was detected using ChamQ Universal SYBR qPCR Master Mix (No. Q711-02, Vazyme) as the quantitative reagent. qPCR assays were performed as previously described [55]. The relative expression level of each gene was calculated using the 2^−ΔΔCT^ algorithm with *GmActin11* as the internal control [72]. The results were then normalized. Three independent replicates were performed for each treatment. The primers used are described in Appendix A.

## 5. Conclusions

In this study, 66 soybean Snf2 family members were identified. These Snf2 family genes are unevenly distributed across all 20 chromosomes. Soybeans are important food and economic crops. Unlike other non-leguminous crops, soybeans can form a symbiotic relationship with rhizobia for biological nitrogen fixation. Nodule specificity *cis*-elements are widely distributed in the promoters of many soybean Snf2 family genes. Gene expression analysis showed that most of the detected genes had significant expression before and after rhizobia inoculation. This indicates the potential role of Snf2 family proteins in regulating symbiotic nodulation and helps future research on soybean Snf2 family proteins.

## Figures and Tables

**Figure 1 ijms-24-07250-f001:**
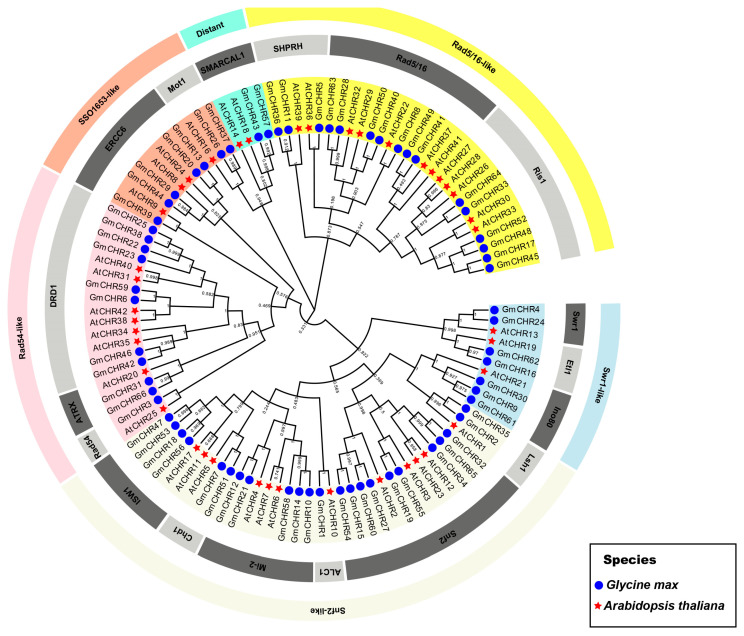
Phylogenetic tree of Snf2 family proteins from soybean and Arabidopsis. The maximum likelihood tree was constructed using MEGAX software with 1000 bootstrap replications. Blue circles and red pentagrams represent Snf2 family proteins from soybean and Arabidopsis, respectively. Different groups of proteins are distinguished using different background colors.

**Figure 2 ijms-24-07250-f002:**
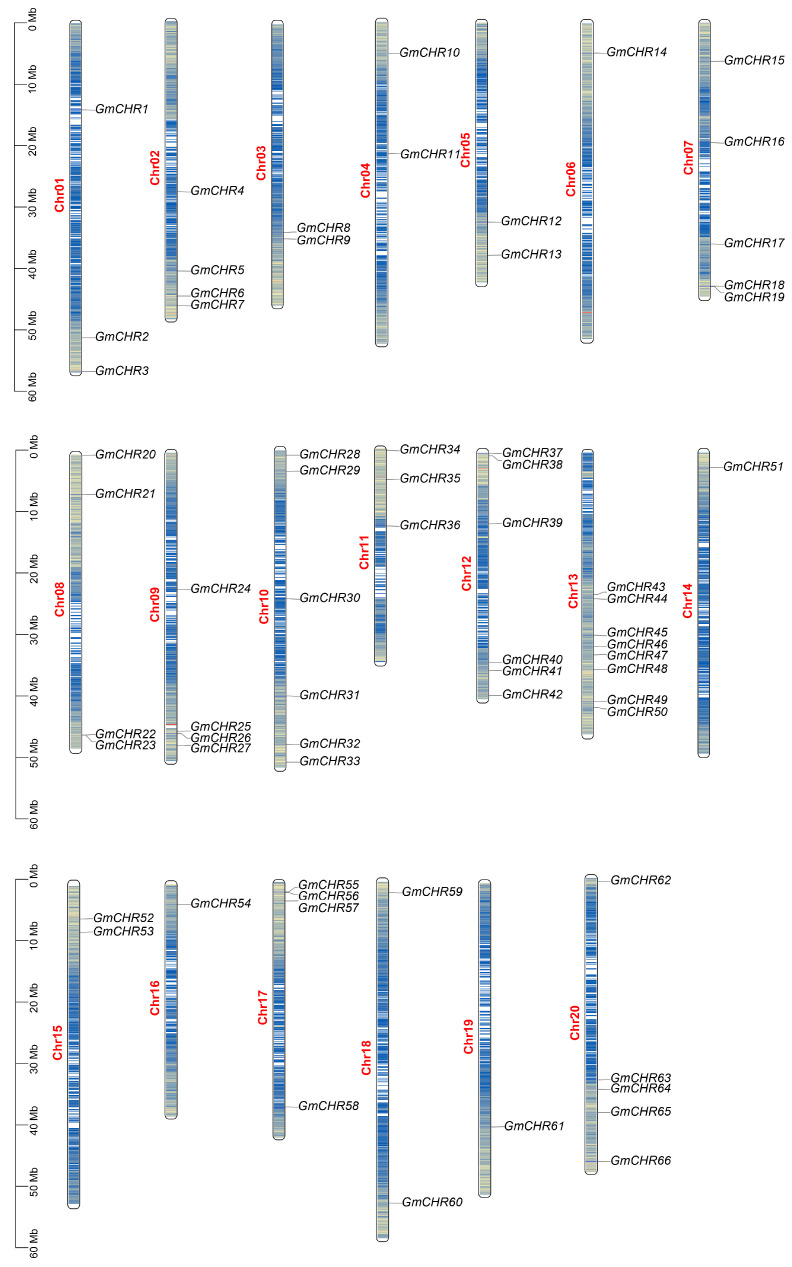
The distribution of soybean Snf2 family genes across chromosomes.

**Figure 3 ijms-24-07250-f003:**
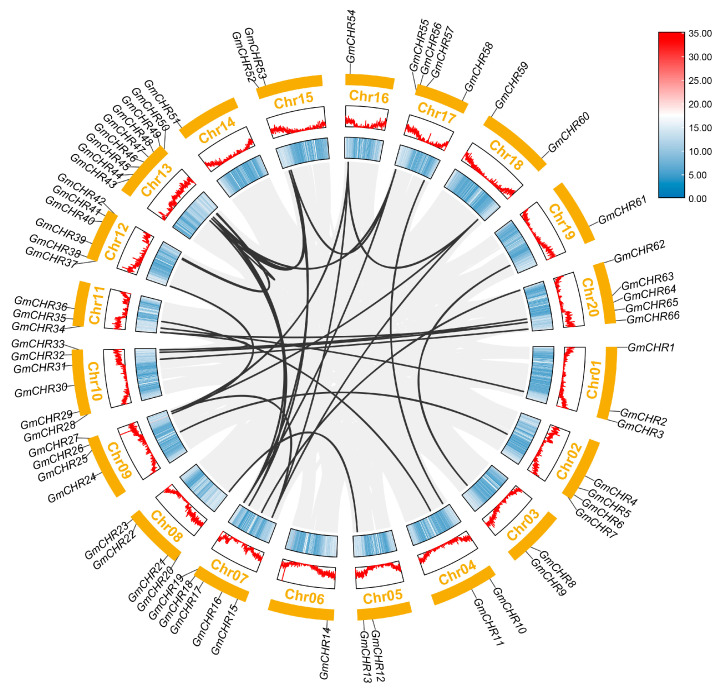
Black curves connect segmentally duplicated gene pairs.

**Figure 4 ijms-24-07250-f004:**
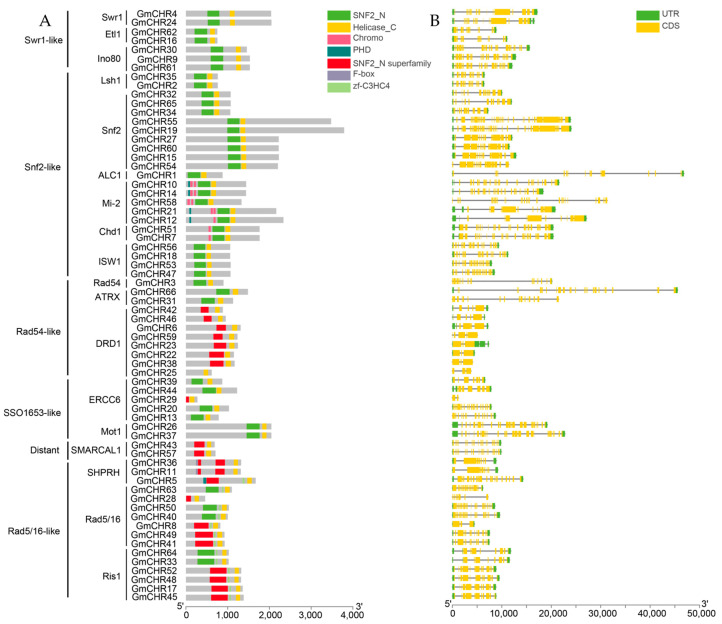
Protein domain analysis of the Snf2 family proteins, as well as the gene structure analysis of the Snf2 family genes. (**A**) Seven conserved motifs of Snf2 family proteins were identified by CDD, shown with different colors representing distinct motifs. At the bottom, the scale bars denote the lengths of the protein sequences. (**B**) The gene structures of Snf2 family genes are illustrated, with yellow boxes representing the coding sequences (CDS), black lines indicating introns, and green boxes depicting untranslated regions (UTR). The scale bars at the bottom indicate the lengths of the genomic sequences.

**Figure 5 ijms-24-07250-f005:**
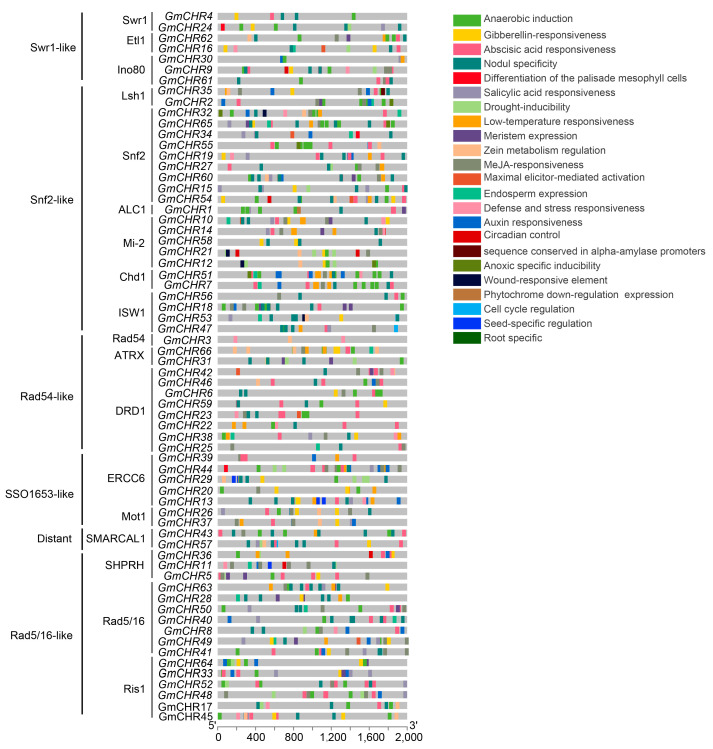
Prediction of *cis*-elements in the promoter regions of Snf2 family genes. Different colored boxes represent different types of *cis*-element. The scale bars at the bottom of the figure indicate the length of the promoter sequence.

**Figure 6 ijms-24-07250-f006:**
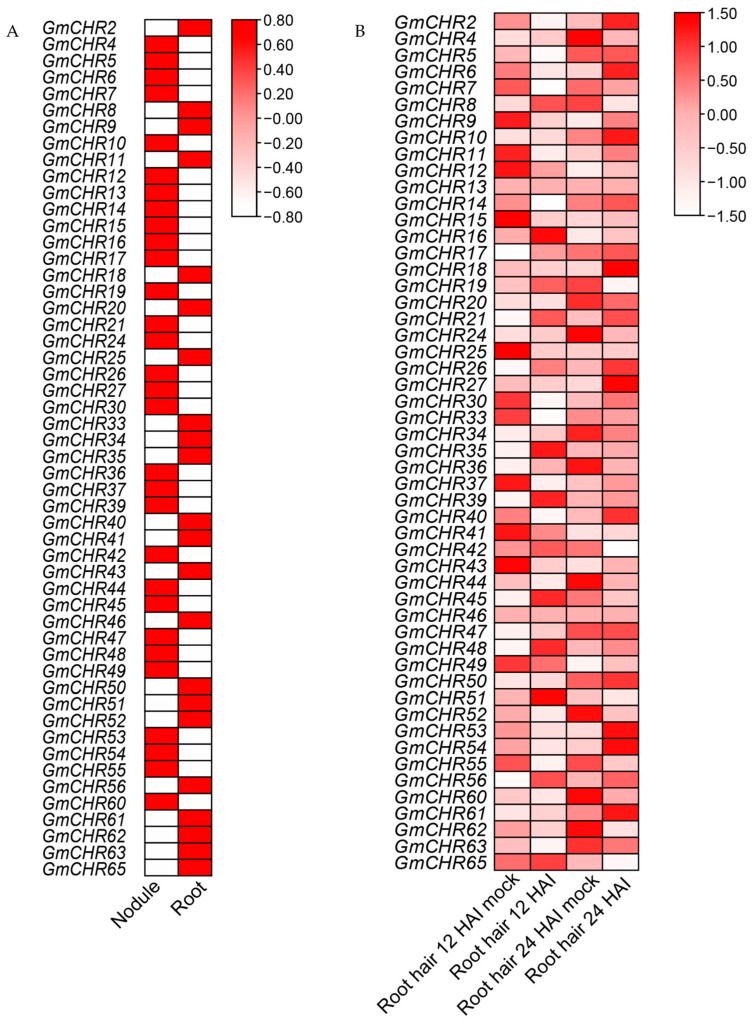
A heat map was generated based on the Reads Per Kilobase of exon model per Million mapped reads (RPKM) values of soybean Snf2 genes to depict the expression patterns of the Snf2 family genes in the root and nodule, as well as its response to rhizobium. (**A**) Heat map illustrating the expression patterns of Snf2 family genes in roots and nodules of soybean. (**B**) Heat map illustrating the expression patterns of Snf2 family genes in root hair at 12/24 h after inoculation (HAI). The expression data of the Snf2 genes were row scale normalized respectively.

**Figure 7 ijms-24-07250-f007:**
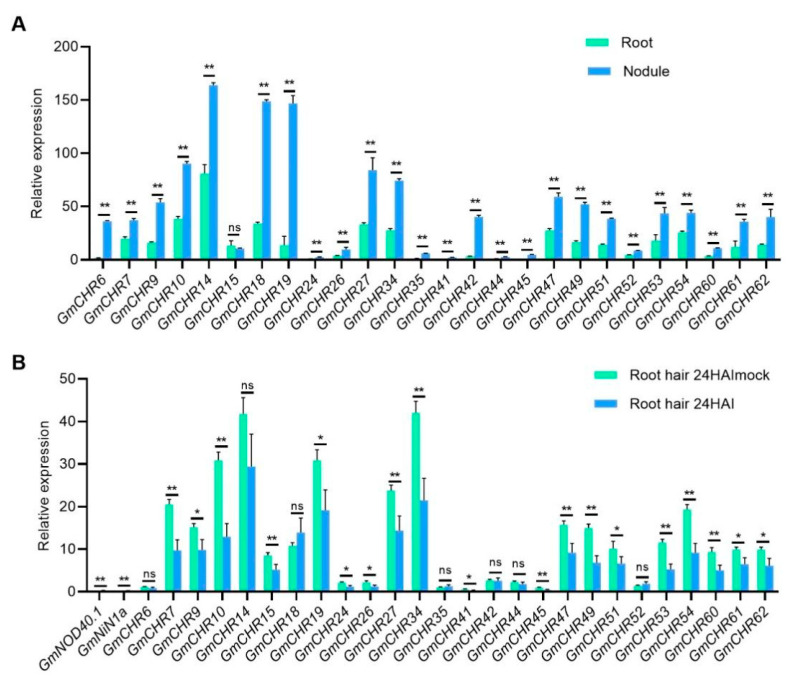
Snf2 gene expression patterns in roots and nodules and their reaction to rhizobium. (**A**) The expression patterns of specific Snf2 genes in roots and nodules were analyzed using qRT-PCR (**B**) Expression patterns of Snf2 genes in root hair were analyzed using qRT-PCR at 12/24 h after rhizobium inoculation (HAI). The color scale ranges from white to red, indicating low or high levels of gene expression. The term ‘mock’ refers to samples without rhizobia inoculation. Data was the most representation of three biological replicates, and the *GmActin11* gene was chosen as the internal reference. A Student’s t-test was applied to assess the significances of the difference between the two groups. * *p* < 0.05. ** *p* < 0.01. “ns” indicates that there is no significant difference.

## Data Availability

The datasets used and/or analyzed during the current study are available from the corresponding author on reasonable request. However, most of the data is shown in Appendix A.

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
