# Peer review of "Genome-Wide Identification and Characterization of the Soybean Snf2 Gene Family and Expression Response to Rhizobia"

_ijms, 2023, doi:10.3390/ijms24087250_

Round 1

Reviewer 1 Report

Authors present a descriptive manuscript about one of the important protein families in the soybean. While the manuscript is well-put together and leaves a pleasant impression, I thought that the concept should be seriously reworked before submission. Also I think that this theme may not be suitable for IJMS. You can consider Plants, for example. 

Most of the manuscript is very descriptive about the snf2 gene family in the soybean. Authors state that they “present a comprehensive analysis of the Snf2 family genes in soybean and demonstrate their essential role in symbiotic nodulation”. However, not one of the experiments showed a direct connection of snf2 genes with the nodulation process. The only experimental part with PCR analysis showed ambiguous results, which didn’t prove anything. 

Presentation of the RT data should also be reconsidered. How did you select genes for RT PCR analysis? In paragraph 4.8. of the materials RT data processing is not described. How did you calculate the results and how did you use data on GmActin11 gene?

Here are some particular notes:

Abstract: Sucrose nonfermenting 2 (Snf2) is the core component … - what Snf2 stands here for? Gene? Protein? 

In the goals of the abstract it is unclear how Snf2 is connected to symbiosis of rhizobia and why it is important.

Is “amplification of genes” is an appropriate term in this case? 

Introduction: Check if all abbreviations are explained

The soybean root secretes flavonoid compounds to induce the synthesis and secretion of Nod factors (NFs) by rhizobia. – interaction of legumes and rhizobia are a little more complex than that

Introduction is nice, but doesn’t explain the connection between chromatin and symbiosis.

The structure of the manuscript, e.g. placement of the figures does not satisfy the journal’s recommendations.

File with supplement data is missing

Author Response

Authors present a descriptive manuscript about one of the important protein families in the soybean. While the manuscript is well-put together and leaves a pleasant impression, I thought that the concept should be seriously reworked before submission. Also I think that this theme may not be suitable for IJMS. You can consider Plants, for example. 

Most of the manuscript is very descriptive about the snf2 gene family in the soybean. Authors state that they “present a comprehensive analysis of the Snf2 family genes in soybean and demonstrate their essential role in symbiotic nodulation”. However, not one of the experiments showed a direct connection of snf2 genes with the nodulation process. The only experimental part with PCR analysis showed ambiguous results, which didn’t prove anything. 

How did you select genes for RT PCR analysis?

Response:Thank you for your question. We selected Snf2 genes containing nodule specificity cis-element and with Reads Per Kilobase per Million mapped reads (RPKM) greater than 2.5 in nodules in the eFP Browser. We have added an explanation of this selection process to the manuscript(line 263-265).

In paragraph 4.8. of the materials RT data processing is not described. How did you calculate the results and how did you use data on GmActin11 gene?

Response:Thank you for your question. The relative expression level of each gene was calculated based on the 2−ΔΔCT algorithm using GmActin11 as the internal control. The results were then normalized. We have made the necessary changes to the manuscript to reflect this(line 436-437).

Abstract: Sucrose nonfermenting 2 (Snf2) is the core component … - what Snf2 stands here for? Gene? Protein? 

Response: Thank you for your question. Snf2 stands for protein there . We have made the necessary changes in the manuscript to reflect this (line 13).

In the goals of the abstract it is unclear how Snf2 is connected to symbiosis of rhizobia and why it is important.

Response:Thank you for your question. We apologize for the lack of clarity in our description. Our goal is to improve our understanding of the potential regulatory roles of soybean Snf2 family genes in the symbiosis of rhizobia. We have made the necessary changes in the manuscript to reflect this(line 33-34).

Is “amplification of genes” is an appropriate term in this case? 

Response:Thank you for your suggestion. We have made the necessary changes in the manuscript (line 24).

Introduction: Check if all abbreviations are explained The soybean root secretes flavonoid compounds to induce the synthesis and secretion of Nod factors (NFs) by rhizobia. – interaction of legumes and rhizobia are a little more complex than that

Response:Thank you for your suggestion. We have checked and made sure that all abbreviations are explained in the manuscript(line 92-104).

Introduction is nice, but doesn’t explain the connection between chromatin and symbiosis

Response:Thank you for your suggestion. Currently, there is very little research on the connection between chromatin and symbiosis. We have only found one relevant paper and have added the relevant content to the manuscript(line 107-111).

The structure of the manuscript, e.g. placement of the figures does not satisfy the journal’s recommendations.

Response:Thank you for your suggestion. We have made the necessary changes in the manuscript.

File with supplement data is missing

Response:Thank you for your comments. We apologize for any confusion caused by the missing supplement data. We have now added the supplement data.

Reviewer 2 Report

This article presented Genome-Wide Identification and Characterization of the Soybean Snf2 Gene Family and Expression Response to Rhizobia. The study is well organized and data is well arranged. The findings would be helpful for future studies. Before recommending this article for publication, there are some shortcomings for that should be resolve.

In introduction add significance of the study.

Also add novelty of the study.

Provide reason why Soybean was selected for Genome wide identification of Snf2.

Provide economic, nutritional and industrial importance of the Soybean by citing relevant study. https://doi.org/10.3390/ijms22179175

Add line numbers, follow the journal format.

Also provide hypothesis of the study in introduction.

Section 4.8 should be cited with relevant study https://doi.org/10.1016/j.indcrop.2022.116090

Provide complete details of promoter analysis.

Expression analysis italicize species name.

Conclusion of the study is missing. It should be added.

Author Response

This article presented Genome-Wide Identification and Characterization of the Soybean Snf2 Gene Family and Expression Response to Rhizobia. The study is well organized and data is well arranged. The findings would be helpful for future studies. Before recommending this article for publication, there are some shortcomings for that should be resolve.

In introduction add significance of the study.Also add novelty of the study.

Response:Thank you for your suggestion. We have made the necessary changes in the manuscript(line 111-116).

Provide reason why Soybean was selected for Genome wide identification of Snf2.

Response:Thank you for your comments. We have added the significance and novelty of the study to the introduction section(line 111-112).

Provide economic, nutritional and industrial importance of the Soybean by citing relevant study. https://doi.org/10.3390/ijms22179175.

Response:Thank you for your suggestion. We have made the necessary changes in the manuscript(line 89-90).

Add line numbers, follow the journal format.

Response:Thank you for your comments. We have added line numbers and followed the journal format as per your suggestion. We hope that it will be satisfactory.

Also provide hypothesis of the study in introduction.

Response:Thank you for your suggestion.  We have revised the introduction section to include the hypothesis of the study as per your suggestion(line 114-115). 

Section 4.8 should be cited with relevant study https://doi.org/10.1016/j.indcrop.2022.116090

Response:Thank you for your comments. We have cited the relevant study in Section 4.8 as per your suggestion. We hope that it will be satisfactory(line 436).

Provide complete details of promoter analysis.

Response:Thank you for your suggestion. We have made the necessary changes in the manuscript(line 214-228).

Expression analysis italicize species name.

Response:Thank you for your comments. We have reviewed the manuscript and found that there are no species names that need to be italicized in that section. However, we have changed the gene names to italics as per your suggestion. We hope that it will be satisfactory.  

Conclusion of the study is missing. It should be added.

Response:Thank you for your feedback. We have added the conclusion section to the manuscript as per your suggestion. We hope that it meets your expectations(line 440-449).

Round 2

Reviewer 1 Report

Authors significantly improved their manuscript after the first rounf of the review, after which it can be accepted to the IJMS. However, there are a couple more remarks to address:

1. Authors should concider adding two versions of the manuscript after the revision: with and without tracked changes to ease the reading process.

2. I'm glad that authors added connection of Snf2 protein family to nodulation via nodule specificity. But they need to add explanation of the "nodule specificity" term in relation to cis-elements.

Author Response

Authors significantly improved their manuscript after the first rounf of the review, after which it can be accepted to the IJMS. However, there are a couple more remarks to address:

1. Authors should concider adding two versions of the manuscript after the revision: with and without tracked changes to ease the reading process.

Response: Thank you for your feedback and suggestions. We have revised the manuscript according to your comments and suggestions. We appreciate your suggestion of providing two versions of the manuscript: one with tracked changes and one without tracked changes to ease the reading process. We will provide both versions of the manuscript in our response.

2. I'm glad that authors added connection of Snf2 protein family to nodulation via nodule specificity. But they need to add explanation of the "nodule specificity" term in relation to cis-elements.

Response: Thank you for your feedback and suggestions. We have revised the manuscript according to your comments and suggestions. We appreciate your suggestion of adding an explanation of the “nodule specificity” term in relation to cis-elements. We have made the relevant modifications in line 267 of our manuscript.